# Fluoroplast Doped by Ag_2_O Nanoparticles as New Repairing Non-Cytotoxic Antibacterial Coating for Meat Industry

**DOI:** 10.3390/ijms24010869

**Published:** 2023-01-03

**Authors:** Sergey V. Gudkov, Ruibin Li, Dmitriy A. Serov, Dmitriy E. Burmistrov, Ilya V. Baimler, Alexey S. Baryshev, Alexander V. Simakin, Oleg V. Uvarov, Maxim E. Astashev, Natalia B. Nefedova, Sergey Y. Smolentsev, Andrey V. Onegov, Mikhail A. Sevostyanov, Alexey G. Kolmakov, Mikhail A. Kaplan, Andrey Drozdov, Eteri R. Tolordava, Anastasia A. Semenova, Andrey B. Lisitsyn, Vasily N. Lednev

**Affiliations:** 1Prokhorov General Physics Institute of the Russian Academy of Sciences, Vavilove St. 38, 119991 Moscow, Russia; 2All-Russia Research Institute of Phytopathology of the Russian Academy of Sciences, Institute St., 5, Big Vyazyomy, 143050 Moscow, Russia; 3Institute of Biology and Biomedicine, Lobachevsky State University of Nizhny Novgorod, 603105 Nizhny Novgorod, Russia; 4School for Radiologic and Interdisciplinary Science, Soochow University, Suzhou 215123, China; 5Institute of Cell Biophysics, Russian Academy of Sciences, Federal Research Center Pushchino Scientific Center for Biological Research of the Russian Academy of Sciences, Institutskaya St., 3, 142290 Pushchino, Russia; 6Federal State Budget Educational Institution of Higher Education Pushchino State Institute of Natural Science, Science Av. 3, 142290 Pushchino, Russia; 7Mari State University, pl. Lenina, 1, 424001 Yoshkar-Ola, Russia; 8A.A. Baikov Institute of Metallurgy and Materials Science (IMET RAS) of the Russian Academy of Sciences, Leninsky Prospect, 49, 119334 Moscow, Russia; 9Institute for Analytical Instrumentation of the Russian Academy of Sciences, Ulitsa Ivana Chernykh, 31–33, lit. A, 198095 St. Petersburg, Russia; 10V. M. Gorbatov Federal Research Center for Food Systems, Russian Academy of Sciences, Talalikhina St., 26, 109316 Moscow, Russia

**Keywords:** silver oxide nanoparticles, polytetrafluoroethylene, foodborne illness, antibiofilm activity, reactive oxygen species, cytotoxicity

## Abstract

Foodborne infections are an important global health problem due to their high prevalence and potential for severe complications. Bacterial contamination of meat during processing at the enterprise can be a source of foodborne infections. Polymeric coatings with antibacterial properties can be applied to prevent bacterial contamination. A composite coating based on fluoroplast and Ag_2_O NPs can serve as such a coating. In present study, we, for the first time, created a composite coating based on fluoroplast and Ag_2_O NPs. Using laser ablation in water, we obtained spherical Ag_2_O NPs with an average size of 45 nm and a ζ-potential of −32 mV. The resulting Ag_2_O NPs at concentrations of 0.001–0.1% were transferred into acetone and mixed with a fluoroplast-based varnish. The developed coating made it possible to completely eliminate damage to a Teflon cutting board. The fluoroplast/Ag_2_O NP coating was free of defects and inhomogeneities at the nano level. The fluoroplast/Ag_2_O NP composite increased the production of ROS (H_2_O_2_, OH radical), 8-oxogualnine in DNA in vitro, and long-lived active forms of proteins. The effect depended on the mass fraction of the added Ag_2_O NPs. The 0.01–0.1% fluoroplast/NP Ag_2_O coating exhibited excellent bacteriostatic and bactericidal properties against both Gram-positive and Gram-negative bacteria but did not affect the viability of eukaryotic cells. The developed PTFE/NP Ag_2_O 0.01–0.1% coating can be used to protect cutting boards from bacterial contamination in the meat processing industry.

## 1. Introduction

Bacterial contamination of meat products can occur at any stage of its journey to the consumer: from slaughter and carcass-cutting to storage at points of sale [1,2]. At the same time, bacteria can form biofilms on work surfaces that are resistant to classical disinfection methods [3,4]. Contamination of meat processing products with bacteria of epidemiological significance (*Listeria monocytogenes*, *Clostridium perfringens*) is a serious global health problem. In total, in worldwide in 1990–2012, approximately 2 billion cases of foodborne infections were registered and caused over 1,000,000 deaths [5]. In the European Union, the number of cases of foodborne infections registered in 2005 exceeded 197,000 cases per year. In 2016, this figure increased to 240,000 cases per year [6,7]. In Canada, the incidence of foodborne infections is comparable [8]. In the United States, the number of cases of foodborne illness (foodborne illness) of a bacterial nature exceeds 48 million cases per year, of which there are 128,000 hospitalizations and over 3000 deaths [9]. In Africa and Asia, foodborne infections are of particular danger, as they are so common among children [10]. Danger to the life and health of consumers is not only due to the microorganisms themselves but also due to the produced or bacterial toxins with various natures [11]. In this regard, it is very important to prevent bacterial contamination during the initial stages of meat preparation in order to minimize the possible accumulation of bacterial toxins. In some cases, foodborne infections can lead to serious complications, as well as death.

Among the complications of bacterial infections were reported lesions of the gastrointestinal tract (gastritis, stomach ulcers, severe forms of diarrhea), CNS (meningitis, encephalitis), kidneys, liver, spleen, musculoskeletal system (reactive arthritis), cardiovascular system (endocarditis), and reproductive system (premature birth, stillbirth) [12,13,14,15,16,17,18].

According to the NCBI, the first studies on bacterial food contamination date back to the first half of the twentieth century. Over the past 60 years, the rate of publications on this subject has grown from a few dozen papers per year to hundreds and even thousands of papers per year. This indicates not only the high significance of the problem but also the lack of a solution and the continuation of active research in this direction.

One of the promising methods for combating the bacterial contamination of work surfaces at food industry enterprises is the use of polymer coatings designed to reduce the adhesion of bacteria to treated surfaces [19,20,21].

One of the most promising polymers for this application is fluoroplasts (Teflon, fluoroplast, polytetrafluoroethylene, PTFE). It has been shown that the adhesion of bacteria to working surfaces coated with fluoroplasts is lower than to other surfaces, in particular, ceramic ones [22]. Fluoroplasts are a polymer consisting of 1,1,2,2-tetrafluoroethylene monomers and can be described by the formula -(CF_2_-CF_2_)_n_-.It is an opaque and smooth milky white material. Fluoroplastics are widely used in various fields: mechanical engineering, chemical production, space and marine industries, biomedicine in the manufacture of implants, and in the food industry [23,24,25,26,27,28,29,30,31]. The breadth of applications of fluoroplastics is due to its unique properties: high mechanical and chemical resistance, thermal and electrical stability, low coefficient of friction, self-lubrication, and self-healing ability [32,33,34,35]. An additional advantage of fluoroplasts is its very low toxicity to mammals in vivo [36]. Due to pronounced hydrophobic properties and mechanical resistance, fluoroplasts can be used to reduce adhesion to working surfaces in food production [32,33,34].

In a number of studies, the antibacterial activity of polymeric materials was enhanced by adding metal NPs and their oxides, for example, Ag, Cu, ZnO, or Ag_2_O [37,38,39,40]. The antimicrobial properties of metal NPs and their oxides are well described in numerous works [41,42,43,44]. Among the general pool of works, since 2005, a significant part (over 30% of publications) are publications devoted to the antibacterial activity of silver NPs, including those in the form of silver oxide. Ag_2_O NPs are able to realize their antibacterial action through several mechanisms: enzyme inactivation, genotoxic action, disruption of cell wall integrity, and photocatalytic action (Table 1).

Based on the foregoing, one of the promising materials in this case is a composite based on fluoroplasts and Ag/Ag_2_O NPs. In particular, such a composite was developed for medical applications [55]. However, we believe that this composite can be used in the protection of work surfaces in the food industry, in particular, the processing of cutting boards in the processing of meat. Unfortunately, new data on the development of bacterial resistance against NPs of metals and metal oxides have recently appeared. Bacterial defense mechanisms include the increased expression of extracellular matrix molecules (flagellin) to inactivate NPs, the release of pigments to inactivate metal ions, and the activation of antioxidant defense to combat oxidative stress [56,57]. It should be noted that these mechanisms are most likely implemented after a certain amount of time after the introduction of single NPs [58]. We assume that the constant dosed release of new portions of NPs from a material during their using can be a way to reduce bacterial resistance against NPs. Composite materials based on polymers and NPs are capable of the controlled release of NPs.

Previously, we showed that Ag_2_O NPs, when added to our previously created composite materials based on PLGA and borosiloxane, showed a good bacteriostatic effect and low cytotoxicity against eukaryotic cells [59,60]. In this work, we, for the first time, attempted to create a nanocomposite coating based on fluoroplasts and Ag_2_O NPs, to study its physicochemical properties, the effect on the generation of ROS and markers of DCH and protein oxidation, bacteriostatic and bactericidal actions, and cytotoxicity against eukaryotic cells.

## 2. Results and Discussion

### 2.1. Physicochemical Properties

At the initial stage of the study, we characterized the main parameters of the obtained NPs: size, ζ-potential, absorption spectrum, and shape (Figure 1). The resulting NPs have a monomodal size distribution with an average hydrodynamic diameter of ~45 nm and a half-width of no more than 35–55 nm (Figure 1a). The resulting size distribution coincides with the literature data [50,75,76,77]. The average value of the ζ-potential of the obtained NPs was ~32 mV, and the half-width was from −42 to 28 mV (Figure 1b). The data obtained agree with the literature data and indicate the stability of the aqueous colloid of the synthesized NPs [78,79]. In the absorption spectrum of colloidal NPs, a clear peak was observed in the region of ~420 nm, which is characteristic of Ag_2_O NPs [80]. According to TEM data, the obtained NPs have a round shape (Figure 1d), which is consistent with the literature data [50]. Thus, by laser ablation in water, we obtained rounded Ag_2_O NPs with a unimodal size distribution that are stable in aqueous colloids.

The resulting Ag_2_O NPs were mixed with a fluoroplast-based varnish (see Section 3).

### 2.2. Composite Material Preparation and Characterization

At the next stage of the study, we tested the ability of the developed coating based on fluoroplastic varnish and Ag_2_O NPs to fill visible damage on a fragment of a used Teflon cutting board (Figure 2a). After drying, the composite coating remained evenly distributed over the treated surface, and no visible damage was observed (Figure 2b). When trying to mechanically damage the surface of the coating, no scratches, chips, or other traces of mechanical impact were observed.

To assess the microrelief of the treated surface, the samples were analyzed using the AFM method. Fluoroplastic without the addition of NPs after drying forms a smooth surface without cracks, protrusions, or other inhomogeneities (Figure 3a). The difference in the height of the sample section with an area of ~1 μm^2^ did not exceed 2–3 nm (Figure 3c).

The addition of Ag2O NPs at a concentration of 0.1% did not affect the coating morphology (Figure 3b). The difference in the height of the sample area with an area of ~1 µm^2^ also did not exceed 2–3 nm (Figure 3d). Using the AFM method, we did not detect NPs on the surface of the samples; therefore, most of the Ag2O NPs are contained in the bulk of the polymer matrix. The obtained data on the high degree of homogeneity of the microrelief of the fluoroplastic coating are consistent with the literature data obtained for the fluoroplastic [81,82]. To estimate the distribution of Ag_2_O NPs inside the polymer coating, we used the MIM method based on the difference between the refractive indices of the NPs and the polymer matrix [83]. In the PTFE without the addition of NPs, we did not find large inhomogeneities (Figure 4a).

After the addition of 0.001% Ag2O NPs, structures were observed in the polymer matrix, presumably being aggregates of NPs. The aggregates had an average length of 1–2 µm and a width of 0.5–1 µm. The phase difference between the NP aggregates and the polymer matrix was no more than 80 nm (Figure 4b). With an increase in the mass fraction of Ag2O NPs to 0.01%, an increase in the length of aggregates up to 3–5 µm was observed. The phase difference values remained comparable (Figure 4c). At an Ag2O NP concentration of 0.1%, the aggregate width increased to 2–3 μm. The aggregate lenght exceeded 8 μm. The phase difference increased to 100 nm and above (Figure 4d). Thus, we can assume that the size of NP aggregates depends on the concentration in the polymer matrix. Our data agree with the literature data on the ability of metal oxide nanoparticles to aggregate in a polymer matrix [84]. Similar data were obtained by us for other nanocomposites using polymer matrices of PLGA and borosiloxane [85,86,87].

### 2.3. Effect of Nanocomposite on ROS Generation

At the next stage, the ability of composite coatings based on PTFE and NPs Ag_2_O to influence the formation of ROS in aqueous solutions was studied (Figure 5). It was found that fluoroplasts without the addition of NPs (“Fluoroplast” group) did not affect the formation of hydroxyl radicals or hydrogen peroxide ([H_2_O_2_] = 3.5 ± 0.4 nM, [•OH] = 23 ± 5 nM). At the same time, the addition of NPs Ag_2_O to the fluoroplast matrix increased the concentration of the formed hydroxyl radicals, even at the minimum concentration of NPs (0.001%). The concentration of H_2_O_2_ in solutions aged with fluoroplast/NPs Ag_2_O composite materials was 5.8 ± 0.8, 11.8 ± 1.9, and 23.5 ± 2.2 nM for composite coatings containing 0.001, 0.01, and 0.1% NPs Ag_2_O, respectively. The concentration of hydroxyl radicals in the fluoroplast functionalized with NPs Ag_2_O was 35 ± 6, 45 ± 7, and 60 ± 8 nM for composite coatings containing 0.001, 0.01, and 0.1% NPs Ag_2_O, respectively.

ROS generation (intracellular and extracellular) is one of the most important and key mechanisms of the antibacterial action of metal NPs and metal oxides, including NPs Ag_2_O. An increase in the formation of ROS, such as superoxide (O_2_^−^), hydroxyl radicals (•OH), and hydrogen peroxide (H_2_O_2_) outside cells under the influence of Ag NPs, including those mediated by Ag^+^, has been reported [88,89,90]. Hydrogen peroxide is one of the most stable ROS that can be transported across the membrane. Hydroxyl radicals are highly reactive but unable to pass through cell membranes. However, they can be sources of secondary radicals [91,92,93]. The generation of moderate amounts of ROS is an important part of the functioning of normal eukaryotic cells: ROS are involved in the regulation of cell division, differentiation, and migration [94]. In the case of the excessive production of ROS and/or the disruption of the functioning of antioxidant systems, the development of “oxidative stress” is possible, leading to DNA modification, protein inactivation, lipid peroxidation, etc. [95,96,97]. “Oxidative stress” of cells may be associated with carcinogenesis, mutagenesis, and accelerated aging processes [98]. The amounts of ROS, both hydrogen peroxide and hydroxyl radical, generated in the presence of composite coatings, do not exceed 25 and 60 nM, respectively (Figure 5). It is known that ROS concentrations below 1 μM are characteristic of normal cell functioning [99]; therefore, ROS generation in the presence of a nanocomposite coating cannot be considered a major mechanism of antimicrobial action.

### 2.4. 8-Oxoguanine and LRPS Generation Study

We have studied the effect of the developed composite coating on the generation of markers of “oxidative stress” and damage to DNA and proteins: 8-oxoguanine and LRPS. The first is formed during the oxidative modification of DNA guanine. LRPS are formed during protein modification, while LRPS can be sources of new secondary free radicals [62,97,100,101]. The PTFE coating without the addition of NPs did not affect the generation of either 8-oxoguanine (Figure 6a) or LRPS (Figure 6b). The addition of Ag_2_O NPs to the fluoroplastic matrix increased the generation of 8-oxoguanine in a dose-dependent manner (Figure 6a).

The addition of 0.001% Ag_2_O NPs increased the generation of 8-oxoguanine by ~50% compared to the control. The addition of 0.01 and 0.1% Ag_2_O NPs increased the generation of 8-oxoguanine by ~100 and ~180% compared to the control. The enhancement of 8-oxoguanine generation in the presence of Ag_2_O NPs agrees with the literature data [51,52]. The addition of Ag_2_O NPs at concentrations of 0.001, 0.01, and 0.1% increased LRPS generation by 20, 90, and 200% compared to control (Figure 6b). It is noteworthy that the half-life rate of LRPS did not depend on the presence of Ag_2_O NPs and their concentration and was 5 h in all variants of the experiment. We have found the potential ability of a composite coating of fluoroplast/Ag_2_O NPs to enhance the oxidative modification of biopolymers using DNA and proteins as an example. The effect is due solely to the properties of NPs Ag_2_O.

### 2.5. Evaluation of Antibacterial Activity and Antibiofilm Activity

We have evaluated the bacteriostatic effect of the developed composite material PTFE/Ag_2_O NPs, as well as the fluoroplastic coating without the addition of Ag_2_O NPs (Figure 7). Fluoroplast without the addition of NPs had no effect on the growth of all studied bacteria. 

The addition of 0.01 or 0.1% Ag_2_O NPs significantly inhibited the growth of all studied bacteria after 6 and 18 h of cultivation. In the case of *L. monocytogenes* on the fluoroplastic/NP Ag_2_O 0.01% composite, the growth rate was reduced by an order of magnitude compared with the control after 6 h and by three orders of magnitude compared with the control after 18 h of cultivation (Figure 7a). The fluoroplast/NP Ag_2_O 0.1% composite had a comparable bacteriostatic effect against *L. monocytogenes*. In the case of *S. aureus*, the addition of 0.01% Ag_2_O NPs showed a more pronounced growth inhibition after 6 h (by 1.5 orders of magnitude compared to the control) and a less pronounced inhibitory effect after 18 h (by 2.5 orders of magnitude compared to the control). Increasing the dopant concentration to 0.1% increased the bacteriostatic effect of the composite. The number of *S. aureus* was reduced by three orders of magnitude compared to the control (Figure 7c). In the case of *P. aeruginosa*, the addition of Ag_2_O at concentrations of 0.01 and 0.1% caused a comparable bacteriostatic effect. There was a decrease in the number of bacteria by 1.5–2 orders of magnitude compared with the control after 6 h and a decrease by 3–3.5 orders of magnitude compared with the control after 18 h (Figure 7b). Composite coating with the addition of Ag_2_O at concentrations of 0.01 and 0.1% also inhibited the growth of *S. typhimurium* by 2.5 orders of magnitude after 6 h and 3.6–4 orders of magnitude (Figure 7d). In this case, the effect was practically independent of the concentration of Ag_2_O NPs.

The data obtained indicate a significant bactericidal effect of the developed composite coating, which inhibited the growth of biofilms of all considered microorganisms. Composite coatings based on fluoroplast containing 0.01 and 0.1 wt % of Ag_2_O NPs reduced the number of CFU by ~three orders of magnitude for all considered bacterial species. It is noteworthy that the bacteriostatic effect depended to a greater extent on the species of bacteria and not on Gram staining. However, similar data were also obtained by other authors [102,103]. The results of the assessment of bacterial viability are shown in Figure 8. We found that there are practically no dead bacterial cells on the surface of the control sample and that the PTFE coating without NPs, the morphology is without disturbances (Figure 8a,b). On a sample of the composite coating with the addition of 0.1% Ag_2_O NPs, the bacterial cell morphology was preserved, but, at the same time, a significant increase in the number of dead cells was observed (Figure 8c). A small number of living cells were found in this sample, but their number relative to the dead is extremely small. In addition, a significant violation of bacterial cell morphology and destruction of the biofilm were found (Figure 8c). 

We have shown a significant bacteriostatic and bactericidal effect of the composite coating against both Gram-negative and Gram-positive bacteria. To date, a large number of studies have been accumulated, demonstrating not only the antibacterial (bacteriostatic and bactericidal) but also the antimycotic effect of silver oxide nanoparticles, as well as composite materials containing Ag_2_O NPs. At least five key mechanisms for the realization of the antibacterial effect of silver nanooxide have been reported [104]: (1) the formation of silver cations that have a destructive effect on the bacterial cell wall [105]; (2) binding to SH-groups of proteins, leading to the disruption of their functional activity [106]; (3) ROS-mediated toxicity [107]; (4) binding to the N7 atom of guanine in DNA, leading to the disruption of the replication process and the suppression of cell division [52]; and (5) photocatalytic activity of Ag_2_O-NPs, which enhances the photocatalytic properties of the NPs of other metals and metal oxides [108]. Of particular interest is the use of Ag_2_O NPs in combination with polymer matrices in the form of composite materials, which makes it possible to improve both the antibacterial properties and the biocompatibility of the material. Table 1 lists studies demonstrating the antibacterial activity of both Ag_2_O NPs in pure form and as part of polymer–NP composite materials.

Notably, 0.1% by weight can be roughly considered as 10 μg/mL. In this case, the antibacterial properties of the fluoroplast/NPs Ag_2_O 0.1% nanocomposite significantly exceed the results of most studies (Table 2) [109,110,111,112].

### 2.6. In Vitro Cytotoxic Study

We evaluated the acute cytotoxicity of the developed composite coating (Figure 9). When cultivating mouse fibroblasts on a fluoroplast/Ag_2_O NP coating, they showed normal morphology, high density, and, in some areas, almost complete confluence (Figure 9a). Neither the PTFE coating nor the PTFE/NP Ag_2_O 0.1% composite had any effect on cell viability (Figure 9b). The number of non-viable cells in all variants of the experiment did not exceed 5%. Cultivation of fibroblasts on a PTFE substrate without Ag_2_O NPs did not affect the culture density (Figure 9c). An almost twofold increase in cell culture density from ~380 cells/mm^2^ to ~700 cells/mm^2^ was observed on the fluoroplastic/NP Ag_2_O 0.1% composite coating. Neither the PTFE nor the PTFE composite with cell/mm^2^ NPs affected the nuclear area of cultured cells (Figure 9d). Consequently, the composite coating based on fluoroplasts and Ag_2_O NPs had almost no effect on cell survival and nucleus size and therefore did not exhibit cytotoxicity against mouse fibroblasts. We have discovered an interesting fact in the increase in the density of a cell culture on a composite material with Ag_2_O NPs. The ability of Ag NPs to accelerate cell proliferation is described in the literature [118]. Probably, the increase in cell culture density on the fluoroplast/NP Ag_2_O composite is due to the presence of NP Ag_2_O at a concentration higher than 2 μg/mL [118].

It should be noted that the data on the cytotoxicity of NPs Ag_2_O and their composites with polymers are rather contradictory. In a number of works, it was noted that the introduction of nanoparticles into the culture medium reduced the viability of HepG2 [110] and A549 [76] cells. One of the ways to reduce the cytotoxicity of Ag_2_O NPs is the use of Ag_2_O-NPs in the composition of polymer matrices and conjugated with polymers (for example, silk fibroin), which significantly reduced cytotoxicity against the 3T3 fibroblast line [112] and the SH-SY5Y cell line [60]. Examples of the successful functionalization of synthetic [115] and natural [112] fibers with NPs Ag_2_O, which have pronounced antibacterial properties, are described, which can be useful in creating tissues and other materials resistant to bacterial growth. Furthermore, the efficient construction of polyethersulfone/cellulose acetate/Ag_2_O-Cu NPs membranes, nanocomposite membranes with antibiofilm activity and capable of removing nitrophenol from aqueous media was noted [115]. Thus, the use of composite materials based on polymers with the addition of NPs Ag_2_O is of interest for the creation of new biosafe antibacterial materials.

## 3. Materials and Methods

### 3.1. Synthesis and Characterization of Ag_2_O-NPs

For the synthesis of silver oxide nanoparticles, the method of laser ablation in deionized water was used. A setup based on a P-Mark TT 100 ytterbium-doped pulsed fiber laser (Pokkels, Moscow Russia) was used. Laser radiation parameters: λ = 1064 nm, τ = 4–200 ns; v = 20 kHz; average power up to 20 W; E = 1 mJ. The layer of liquid (0.05 M NaNO_3_ aqueous solution) above the target (silver plate of high chemical purity) was about 1 mm. Irradiation time varied within 5–20 min. Using the Zetasizer Ultra Red Label (Malvern Panalytical Ltd., Malvern, UK), the hydrodynamic diameter (DLS) and zeta potential (ELS) of the obtained NPs were determined. The woofer diameter was confirmed using a CPS 24,000 (CPS Instruments, Prairieville, LA, USA). Morphological features (shape, topology) of nanoparticles were assessed using a Libra 200 FE HR transmission electron microscope (TEM) (Carl Zeiss, Jena, Germany), and the elemental composition of NPs was determined using energy-dispersive X-ray spectrometry (EDS) using a JED-2300 system. (Carl Zeiss, Jena, Germany). The composition of the obtained colloidal solutions of NPs was confirmed using Cintra 4040 (GBC Scientific Equipment, Braeside, Australia). Samples were prepared for TEM according to the protocol, according to which droplets of colloidal solutions of Ag_2_O NPs were deposited on the surface of a gold mesh (Ø4), dried at room temperature, and evacuated. The sample grid was placed in a titanium holder. More detailed details of the methods are described in previous works [61,62,63].

### 3.2. Composite Material Preparation and Characterization

After the synthesis of nanoparticles, water was replaced with acetone by centrifugation. The colloidal solution of nanoparticles was centrifuged in a Sigma 3-16KL centrifuge (Sigma Laborzentrifugen GmbH, Osterode am Harz, Germany) with a 12,158 rotor for 40 min at 7000× *g*.; the supernatant (water) was replaced with acetone. These manipulations were carried out at least three times. The resulting colloidal solution was mixed with fluoroplastic varnish (Plast Polymer-Prom, Saint-Petersburg, Russia) to a final concentration of nanoparticles of 0.1, 0.01, and 0.001%. The varnish is a fluoroplastic dissolved in a mixture of acetone, butyl acetate, cyclohexanone, and toluene in a ratio of 25:40:10:25 mass parts. To assess the ability of the obtained composite coating to cover visible damage, the section of the Teflon cutting board with damage as a result of the operation was treated with a composite coating and dried for 24 h at room temperature. For experiments on the evaluation of the generation of ROS, 8-oxoguanine, and long-lived active forms of proteins, drops of a solution of nanoparticles in lacquer with a volume of 500 μL were applied to round degreased glasses 25 mm in diameter. Before the start of the experiments, the coatings were dried for 48 h in a fume hood. For cytotoxic and microbiological studies, samples with a composite coating were preliminarily disinfected by soaking in 70% ethanol for 2–3 h. For microbiological studies, varnish was applied to bulk samples of fluoroplasts (4 × 4 × 6 mm). The surface of the composite coatings was evaluated by atomic force microscopy (AFM) using NPX200 (Seiko Instruments, Tokyo, Japan). The distribution of nanoparticles in the composite polymer–polymer NP coating was assessed using a modulation interference microscope (MIM) using a MIM-321 (Amphora Lab, Moscow, Russia) [64].

### 3.3. Quantification of ROS Concentration

The concentration of hydrogen peroxide formed in aqueous solutions was estimated from the intensity of chemiluminescence of the luminol-p-iodophenol-horseradish peroxidase system. Chemiluminescence was measured on a highly sensitive Biotoks-7A-UZE chemiluminometer (Engineering Center-Ecology, Moscow, Russia). Samples of the studied composite material were placed in polypropylene vials (Beckman, Brea, CA, USA) with the addition of 1 mL of “counting solution” prepared immediately before measurement, containing 1 mM Tris-HCl buffer pH 8.5, 50 μM p-iodophenol, 50 µM luminol, and nM horseradish peroxidase 10. The sensitivity of this method is <1 nM. Samples of composite materials containing various concentrations of NPs in the composition (0.001–0.1 wt %) in the form of films 10 mm × 10 mm in size and 200 μm thick were placed in polypropylene vials (Beckman, CA, USA). After incubation in 20 mL of water, 1 mL of a previously prepared “counting solution” was added to the sample.

To quantify the content of hydroxyl radicals in aqueous solutions, the reaction with coumarin-3-carboxylic acid (CCA) was used. A total of 0.2 M PBS (pH 7.4) was added to a solution of CCA in water (0.5 mM, pH 3.6). Coating samples containing various concentrations of NPs in the composition (0.001–0.1 wt %) were added to vials in the form of films 10 × 10 mm in size and 200 μm thick. In the “Control” group, the experiment was carried out without a sample. Next, polypropylene vials with samples and reagents were heated in a thermostat at a temperature of 80.0 ± 0.1 °C for 2 h. As a result of the hydroxylation reaction, a fluorescent product, 7-hydroxycoumarin-3-carboxylic acid (7-OH-CCA), is formed. 7-OH-CCA fluorescence was detected using a JASCO 8300 spectrofluorimeter (JASCO, Tokyo, Japan) at λ_ex_ = 400 nm, λ_em_ = 450 nm [63,65].

### 3.4. Measurement of the Concentration of the Formed Active Long-Lived Forms of Proteins

The number of long-lived reactive proteins (LRPS) was estimated from the chemiluminescence of protein solutions after heating to 40 °C for 2 h. All samples were stored in the dark at room temperature for 30 min after exposure. The measurements were carried out in 20 mL polypropylene vials (Beckman, Brea, CA, USA) in the dark at room temperature on a highly sensitive Biotox-7A chemiluminometer (Engineering Center—Ecology, Moscow, Russia). Protein solutions (BSA) not subjected to heating were used as controls [66].

### 3.5. Quantitative Determination of 8-Oxoguanine in DNA In Vitro by ELISA Method

To quantify 8-oxoguanine in DNA, a non-competitive enzyme-linked immunosorbent assay (ELISA) was used, using monoclonal antibodies specific for 8-oxoguanine (anti-8-OG antibodies). DNA samples (350 μg/mL) were denatured by boiling in a water bath for 5 min and cooled on ice for 3–4 min. Aliquots (42 μL) were applied to the bottom of the wells of the ELISA plates. The DNA was immobilized using a simple adsorption procedure with incubation for 3 h at 80 °C until the solution became completely dry. Non-specific adsorption sites were blocked with 300 μL of a solution containing 1% skimmed milk powder in 0.15 M Tris-HCl buffer, pH 8.7 and 0.15 M NaCl. Next, the plates were incubated at room temperature overnight (14–18 h). The formation of an antigen–antibody complex with antibodies against anti-8-OG (at a dilution of 1:2000) was carried out in a blocking solution (100 µL/well) by incubation for 3 h at 37 °C. Then, it was washed twice (300 µL/well) with 50 mM Tris-HCl buffer (pH 8.7) and 0.15 M NaCl with 0.1% Triton X-100 after 20 min incubation. Next, a complex with a conjugate (anti-mouse immunoglobulin labeled with horseradish peroxidase (1:1000) was formed by incubation for 1.5 h at 37 °C in a blocking solution (80 µL/well). Then the wells were washed 3 times as described above. Next, a chromogenic substrate containing 18.2 mM ABTS and hydrogen peroxide (2.6 mM) in 75 mM citrate buffer, pH 4.2 (100 µL/well), was added to each well. Reactions were stopped by adding an equal volume of 1.5 mM NaN_3_ in 0.1 M citrate buffer (pH 4.3) upon reaching the color. The optical density of the samples was measured on a plate photometer Titertek Multiscan, (Titretek, Helsinki, Finland) at λ = 405 nm [67].

### 3.6. Evaluation of Antibacterial Activity and Antibiofilm Activity of Samples

The antibacterial properties of PTFE/Ag_2_O-NPs coatings containing various concentrations of Ag_2_O-NPs were tested against two Gram-positive *L. monocytogenes* (azithromycin-, erythromycin-, and sulfamethoxazole-resistant), *S. aureus* and two Gram-negative bacterial species, *P. aeruginosa*, *S. enterica* serotype Typhimurium (azithromycin-resistant). Bacterial cultures were obtained from the working collection of the Laboratory of Microbiology of the Research Institute of Food Systems named after V.I. Gorbatov. These strains have previously been isolated from samples of meat products and from work surfaces at meat processing plants. There strains have high epidemiological significance [68,69,70,71]. Luria–Bertani (LB) medium (BD Difco, Franklin Lakes, NJ, USA) and tryptone soy broth (TSB) (Panreac AppliChem, Barcelona, Spain) were used as culture media. As test surfaces for cultivating bacterial cells, we used Teflon cubes with sides of 4 mm × 4 mm × 6 mm, coated with a composite material with different concentrations of Ag_2_O NPs (0.001–0.1%) in the composition, as well as uncoated and with a fluoroplastic coating not containing Ag_2_O NPs. For the study, daily broth cultures of the studied microorganisms with initial concentration 10^9^ CFU/mL were diluted 100 times to a final concentration of 10^7^ CFU/mL in sterile LB broth and poured into sterile test tubes (V = 2 mL). Next, pre-sterilized Teflon cubes (one sample each) were added to each tube and incubated for 6 and 18 h in a thermostat at 37 °C. After incubation, the cubes were washed once with distilled water (to remove planktonic cells), transferred into test tubes with sterile saline (0.9% NaCl solution), and vigorously shaken for 15 min at least 3 times. Further, the obtained washings were titrated (ten-fold dilutions were carried out), transferred to Petri dishes with a = LB agar, and evenly distributed over the surface with a sterile spatula. The results were recorded by counting the number of colony-forming units (CFU) 24 h after incubation at 37 °C.

To study the antibiofilm activity of fluoroplast/Ag_2_O-NP coatings, a broth culture of bacterial cells (V = 30 μL) was applied to the surface of Teflon cubes and left to dry at room temperature for 30 min. Next, to visualize live and dead cells, the samples were stained with a set of fluorescent dyes Filmtracer Live/Dead Biofilm Viability Kit (Invitrogen, Waltham, MA, USA) and analyzed under a microscope with appropriate filters. The kit used contains SYTO^®^9 fluorescent dyes and propidium iodide (PI). Both of them stain the DNA of microorganisms; however, SYTO^®^9 can quickly penetrate the membrane of living bacteria, while propidium iodide (PI) has more difficulty penetrating the walls of living bacteria. After 20 min of staining, living cells are stained green, and dead cells are stained red. Microscopy was performed using the Eclipse Ni (Nikon, Tokyo, Japan) imaging system.

### 3.7. Isolation and Cultivation of Fibroblasts from Mouse Lungs

All manipulations with animal tissues and cells were performed in clean rooms using a Laminar-S class II biological safety cabinet (Lamsystems, Miass, Russia). Primary cell cultures of isolated mouse lung fibroblasts were obtained according to the standard protocol with minor modifications. Male BALC/b 2–3-month-old mice were used in a cytotoxicity assay. Euthanasia of mice was performed by displacing the cervical vertebrae. Using surgical scissors, the lungs were removed from the animal’s thorax. The lungs were placed in a sterile Ø60 Petri dish (TPP, Trasadingen, Switzerland) containing a small volume of PBS solution. The organs were chopped with sterile scissors into pieces with a volume of ~1 mm^3^. Pieces of lung tissue were incubated for 1 h in 25 mL of DMEM medium containing 0.2% type II collagenase at 37 °C on an MR-1 rocking shaker (Biosan, Riga, Latvia). Collagenase was inhibited by 20% FBS. Tissue pieces incubated in collagenase solution were resuspended by pipetting and then passed through a 70 μm EASTstrainer™ sieve (Greiner bio-one, Kremsmunster, Austria). Cells were washed by centrifugation twice at 350× *g* for 5 min in DMEM. The isolated cells were further cultivated in TC T-25 culture mats (TPP, Trasadingen, Switzerland) in DMEM/F12 medium supplemented with 10% FBS, 2 mM L-glutamine, 100 U/mL penicillin, 100 μg/mL streptomycin obtained from PanEco (PanEco, Moscow, Russia). Upon reaching 80–90% confluence, the cells were attached with 0.05% trypsin-EDTA solution (PanEco, Moscow, Russia) for 5 min at 37 °C. Trypsin was inactivated with 10% FBS. Cells were passaged at least 3 times before starting cytotoxic studies [72].

### 3.8. In Vitro Cytotoxic Studies Using Mouse Lung Fibroblast Cultures

Round glasses for microscopy Ø25 Menzel Glaser (Thermo Fisher Scientific, Waltham, MA, USA) coated with PTFE/Ag_2_O-NPs and without (control group) composite coating (V = 500 µL) were sterilized in 70% ethanol for 2–3 h and then placed in the wells of a 6-well plate (TPP, Trasadingen, Switzerland). A cell suspension (50 μL) was applied to each coating sample, then incubated for 45 min for cell adhesion, after which it was brought to a final volume of 1 mL with warm nutrient medium. The total time from the moment the cells were planted on the surface to microscopic measurements was at least 72 h. Cultivation was carried out in an S-Bt Smart Biotherm CO_2_ incubator (Biosan, Riga, Latvia) at 37 °C and 5% CO_2_. The medium for cell culture was DMEM/F12 medium with additives, the preparation of which was described in detail earlier. The initial number of cells in the suspension placed on the surface of the material was ~50,000 cells per sample. Cytotoxicity was assessed using Hoechst 33342 dyes and propidium iodide (PI) (Thermo Fisher Scientific, Waltham, MA, USA). Immediately after incubation, a sample of the composite coating with cultured cells was placed in a quick-change imaging chamber (RC-40LP, Warner Instruments, Holliston, MA, USA), washed thoroughly with PBS, stained with Hoechst 33342 at a concentration of 5 µg/mL, and incubated for 30 min at 37 °C. The sample was then washed with PBS and stained with 2 μM PI (Thermo Fisher, Waltham, MA, USA) for 1 min. Samples were analyzed using a DMI4000 B fluorescence microscope (Leica Microsystems, Wetzlar, Germany) equipped with an SDU-285 digital camera (SpetsTeleTekhnika, Moscow, Russia). Fluorescence spectra were recorded at excitation/emission wavelengths: 350/470 for Hoechst 33342 (D-filter, Leica Microsystems, Wetzlar, Germany) and 540/590 for PI (TRITC filter cube, Leica Microsystems, Wetzlar, Germany). Light-emitting diodes (LED) M375D2m, M490D3 (Thorlabs, Newton, NJ, USA) and white LED (Cree Inc., Durham, NC, USA) were used as light sources for the excitation of Hoechst 33342 and fluorescence of PI. All images were taken at the following LED currents: 100 mA for the M375D2m LED (Hoechst 33342) and 250 mA for the white LED (PI). The exposure time in all experiments was the same: 500 ms for Hoechst 33342 and 700 ms for PI. The detector gain was ×423 and was the same for all fluorophores and experimental conditions [73,74].

Data acquisition and microscope adjustment control were performed using the Win-FluorXE software version 3.8.7. 8-12-16 (J. Dempster, Strathclyde Electrophysiology Software, University of Strathclyde, Glasgow, UK). The data was collected as 12-bit grayscale images. Subsequent analysis was performed using ImageJ2 (Fiji) software version 2.3.1 (NIH, Bethesda, MD, USA). For each variant of the experiment, at least five samples were analyzed. At least 200 cells were analyzed in each sample. ROIs were determined using ImageJ’s “Threshold” and “Particle Analysis” automated procedures. For an image of 1392 × 1024 pixels obtained at a value of ×20, the following parameters were used: “size” = 100–750 and “circularity” = 0.10–1.00. The nuclei had different fluorescence intensities from Hoechst 33342 and PI. To ensure that all nuclei are included in the analysis, we performed a series of thresholding and particle analysis procedures for each image. The images were converted to 8-bit prior to determining the ROI. Threshold levels ranged from 5 to 255 a.u. with a step of 5 c.u. ROIs were saved as binary masks, and then all were combined, with the removal of duplicates. All procedures were combined in automated macros.

## 4. Conclusions

We have successfully developed a new composite coating based on fluoroplasts and silver oxide NPs. The developed coating effectively covers visible damage on the surface of Teflon, which was in operation. At the micro- and nanolevels, the surface of the composite is smooth, without cracks and defects. The resulting material enhances the generation of ROS (hydrogen peroxide and hydroxyl radical), as well as 8-oxoguanine and LRPS. The intensity of the effect depends on the concentration of added Ag_2_O NPs. The resulting fluoroplast/Ag_2_O NP composite demonstrates a strong bacteriostatic and bactericidal effect. Fluoroplast/NP Ag_2_O 0.01–0.1% inhibits the growth of both Gram-positive and Gram-negative bacteria. At the same time, the developed composite did not show cytotoxicity against the primary culture of mouse fibroblasts. The nanocomposite material fluoroplast/NP Ag_2_O can be used to create protective coatings with antimicrobial properties for cutting boards at meat processing plants.

## Figures and Tables

**Figure 1 ijms-24-00869-f001:**
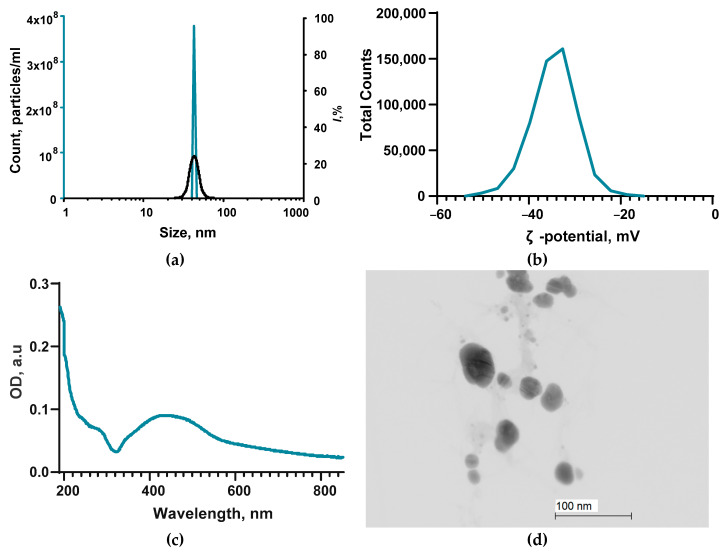
Main characteristics of Ag_2_O NPs: (**a**) Size distribution of NPs obtained by DLS (blue line) and CPS (black line). (**b**) NP distribution over the ζ-potential obtained by the ELS method. (**c**) Absorption spectrum from the UV–vis region of the NP colloid. (**d**) TEM photograph of the obtained NPs. Scale bar: 100 nm.

**Figure 2 ijms-24-00869-f002:**
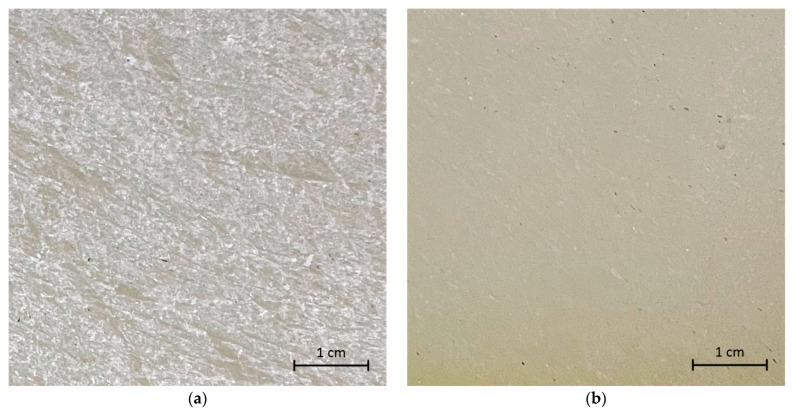
Photographs of a section of a Teflon (PTFE) cutting board with damage before coating (**a**) and after the application and drying of the PTFE/Ag_2_O-NPs 0.1% composite coating (**b**).

**Figure 3 ijms-24-00869-f003:**
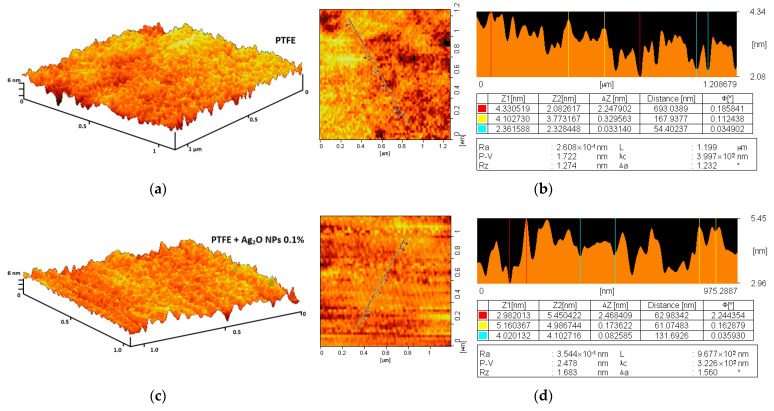
Investigation of the microrelief of a composite material using the AFM method: 3D reconstruction of the surface of a PTFE coating without NPs (**a**) and with the addition of 0.1% Ag_2_O NPs (**c**); examples of the results of a quantitative assessment of the surface inhomogeneity of a PTFE coating without NPs (**b**) and with the addition of 0.1% Ag_2_O NPs (**d**).

**Figure 4 ijms-24-00869-f004:**
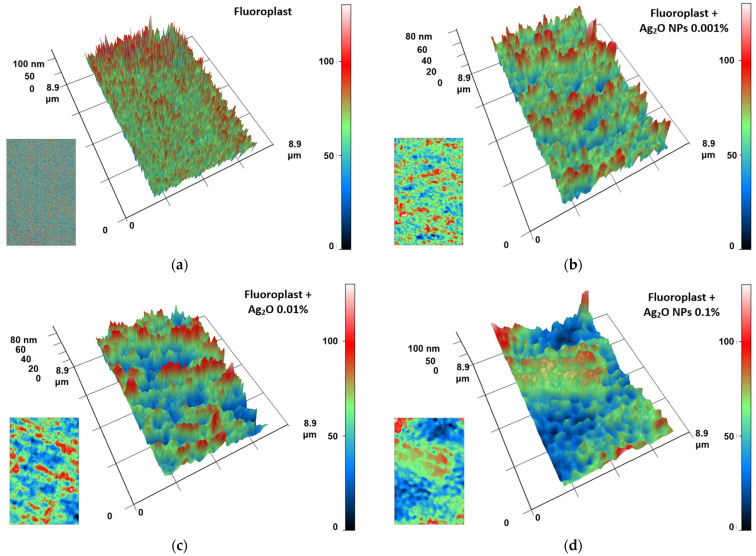
The results of the analysis of the fluoroplast/Ag_2_O NPs composite coating by MIM method: (**a**) PTFE without the addition of nanoparticles; (**b**) PTFE with the addition of 0.001% Fe_2_O_3_ nanoparticles; (**c**) PTFE with the addition of 0.01% Fe_2_O_3_ nanoparticles; (**d**) PTFE with the addition of 0.1% Fe_2_O_3_ nanoparticles. The images are presented as 3D reconstructions, where the abscissa and ordinate axes correspond to the real distance in μm. The Oz axis displays the phase difference in nm (the larger the phase difference, the higher the value on the Oz axis). Coloring is a pseudo color. The initial data on the spatial distribution of the phase difference in the analyzed sample, used to construct 3D reconstructions, are shown in the lower left corners of each panel.

**Figure 5 ijms-24-00869-f005:**
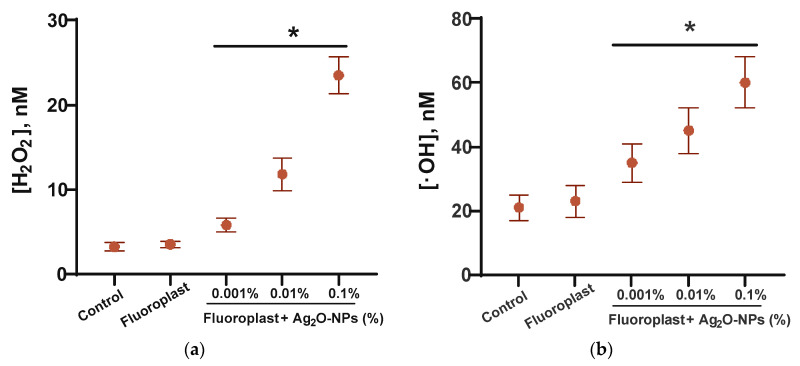
ROS generation in the presence of fluoroplast/Ag_2_O NPs composite coating: (**a**) generation of hydrogen peroxide (2 h, 40 °C); (**b**) generation of hydroxyl radicals (2 h, 80 °C). Data are presented as Mean ± SE. * *p* < 0.05, Mann–Whitney test (*n* = 3).

**Figure 6 ijms-24-00869-f006:**
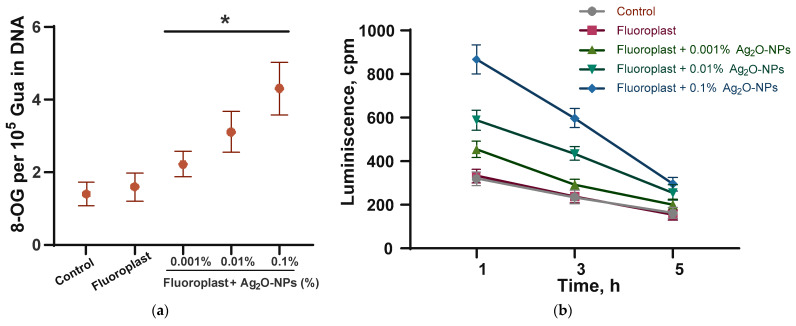
Generation of DNA and protein oxidative damage markers in the presence of fluoroplast/Ag_2_O NPs composite coating: (**a**) generation of 8-oxoguanine in DNA in vitro (2 h, 45 °C); (**b**) LRPS generation (2 h, 40 °C). Data are presented as mean ± SE. * *p* < 0.05, Mann–Whitney test (*n* = 3). 8-OG-8-oxoguanine.

**Figure 7 ijms-24-00869-f007:**
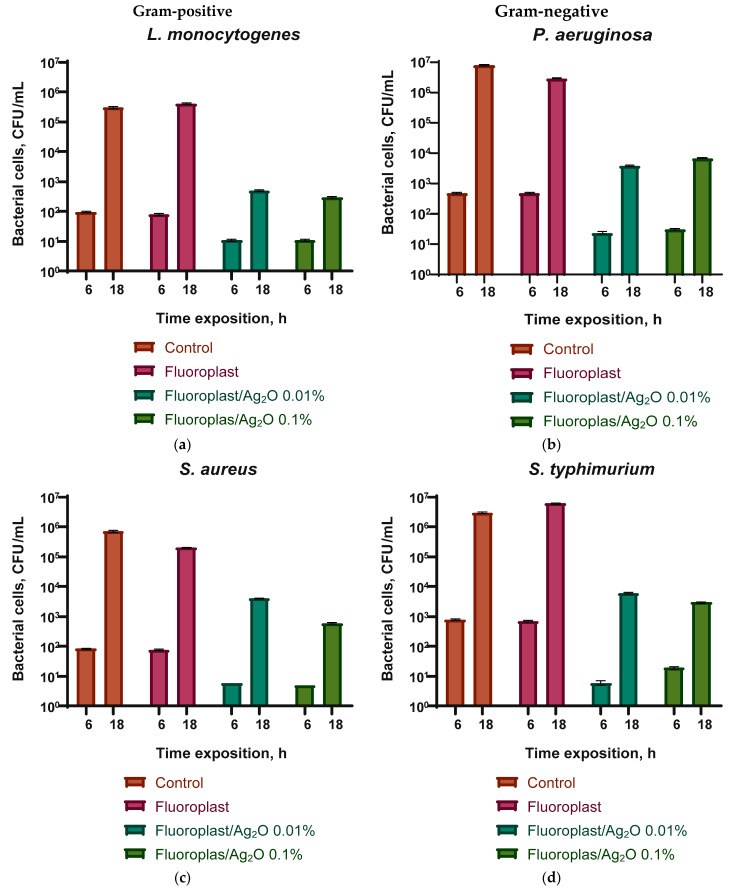
Evaluation of the bacteriostatic effect of a composite material based on fluoroplasts and Ag_2_O NPs against Gram-positive bacteria *Listeria monocytogenes* (**a**) and *Staphylococcus aureus* (**c**) and Gram-negative *Pseudomonas aeruginosa* (**b**) and *Salmonella typhimurium* (**d**) after 6 and 18 h of cultivation. Results are presented as mean ± SE (*n* = 3).

**Figure 8 ijms-24-00869-f008:**
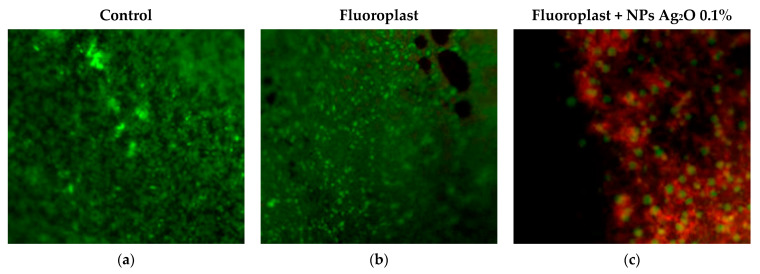
Evaluation of the survival of *P. aeruginosa* on uncoated samples (**a**), on PTFE without NPs (**b**), and on the composite coating of fluoroplast/NP Ag_2_O 0.1% (**c**). Living cells are colored green, dead cells red. Magnification ×400.

**Figure 9 ijms-24-00869-f009:**
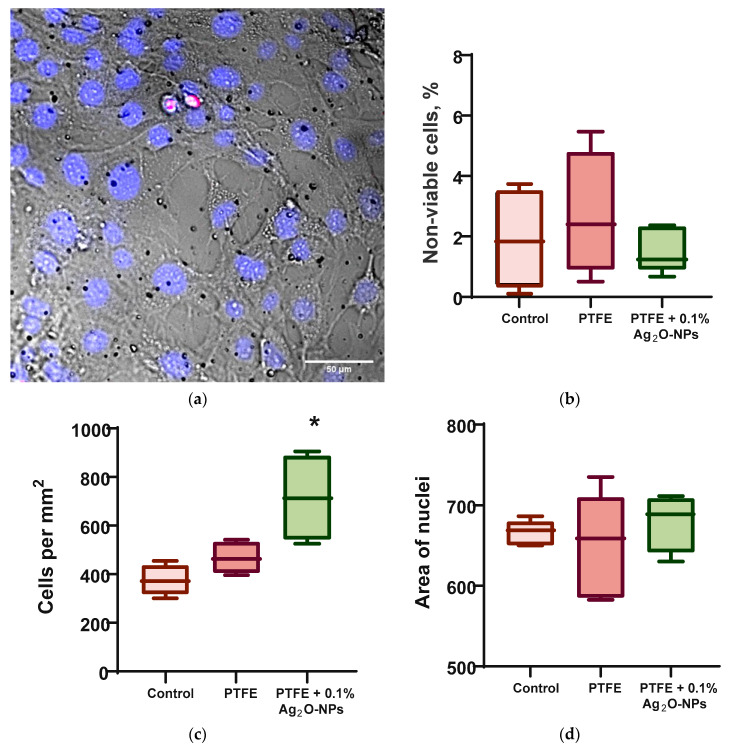
Influence of composite coating fluoroplast/Ag_2_O NPs on the growth of mouse fibroblasts in vitro after 72 h of cultivation. (**a**) An example of a photomicrograph of cells growing on the surface of a composite coating of fluoroplast/NP Ag_2_O 0.1% (merge: blue-Hoechst, red-PI, transmitted light-gray). (**b**) proportion of non-viable cells. (**c**) Density of cell cultures. (**d**) Mean nuclear area. *: statistically significant difference compared to the “Control” group (*p* < 0.05). Data are presented as mean ± standard error of the mean.

**Table 1 ijms-24-00869-t001:** Mechanisms of antimicrobial action of Ag_2_O/Ag NPs.

№	Agent	Mechanism	Ref
1	Ag^+^	Binding to the SH-groups of enzymes, leading to their inactivation. Violation of the functioning of respiratory chain enzymes, accumulation of ROS in the cell.	[45,46,47,48]
2	ROS	Oxidative stress: genotoxic effect, modification of bacterial proteins in DNA (genotoxic effect)	[49,50]
3	Whole NPs	Direct binding to bacterial cell walls and their destruction	[47,48]
4	Ag^+^	Genotoxic effect due to binding to DNA	[51,52]
5	Ag/Ag_2_O *hν*	Photocatalysis	[53,54]

**Table 2 ijms-24-00869-t002:** Literature data on the antibacterial activity of Ag_2_O NPs and nanocomposites based on them.

№	Material Composition	Size (nm); Shape of ZnO-NPs	Microorganism	Concentration	Effect	Ref.
1	Ag_2_O-NPs	~170, nanospheres	*S. aureus*	20–5000 µg/mL	Bactericidal	[109]
2	Ag_2_O-NPs	110–120, nanospheres	*A. flavus*, *A. niger*, *B. subtilis*, *C. albicans*, *E. coli*, *F. solani*, *K. pneumonia*, *M. racemosus*, *P. aeruginosa*, *S. aureus*	28.125–112.5 µg/mL	Bacteriostatic Fungistatic	[110]
3	Ag_2_O-NPs	17.45	*B. aerius*, *B. circulans*, *E. coli*, *P. aeruginosa*	5–7.5 µg/mL	Bacteriostatic Bactericidal	[113]
4	Chitosan/Ag_2_O NPs suspension	10–20, nanospheres	*E. coli*, *S. aureus*	2 µg/mL	Bacteriostatic	[114]
5	Polyethylene terephthalate (PET)/Ag_2_O NPs composite	50–500, rods, nanospheres	*E. coli*		Bacteriostatic	[115]
6	Ag_2_O NPs conjugated with starch in different proportions	30–110, spherical, faceted	*B. cereus*, *E. coli*, *L. onocytogenes*, *P. vulgaris*, *P. putida*, *S. typhymurium*, *S. aureus*, *S. saprophyticus*	100 µg/mL	Bacteriostatic	[111]
7	Ag_2_O NPs conjugated with silk fibroin	15 Nanospheres	*E. coli*, *M. tuberculosis*, *S. aureus*	115.9 µg/mL	Bacteriostatic	[112]
8	Ag_2_O NPs mixed with chitosan solution	~5	*E. coli*, *S. aureus*, *B. subtilis*, *P. aeruginosa*	~5.8 mg/mL	Bacteriostatic	[116]
9	Polyethersulfone (PES)/cellulose acetate (CA)/Ag_2_O NPs nanocomposite and Cu·PES/CA/Ag_2_O NP membranes	20–100	*E. coli*	8 mg/mL	Bacteriostatic	[117]
10	PLGA/Ag_2_O NPs nanocomposite	35 Nanospheres	*E. coli*	10 µg/mL	Bacteriostatic	[60]
11	Fluoroplast/Ag_2_O NPs nanocomposite	45 nanospheres	*L. monocytogenes*, *S. aureus*, *P. aeruginosa*, *S. typhimurium*	10 µg/mL	Bacteriostatic Bactericidal	Present study

## Data Availability

The raw data supporting the conclusions of this article will be made available by the authors, without undue reservation.

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
