# Peer review of "Fluoroplast Doped by Ag_2_O Nanoparticles as New Repairing Non-Cytotoxic Antibacterial Coating for Meat Industry"

_ijms, 2023, doi:10.3390/ijms24010869_

Round 1

Reviewer 1 Report

The authors are congratulated for presenting a well structured manuscript that deals with a very actual subject.

Nevertheless, some comments/suggestions are in order:

- According to the data that the authors present in figure 1, the number of publications concerning antibacterial nanoparticles and antibacterial silver nanoparticles is diminishing. That trend has a reason, which is mainly associated with the development of bacterial resistance to metallic ions and nanoparticles, similarly to what happens with antibiotics. This aspect must be addressed by the authors in the introduction.

- Due to the high number of publications on the general subject of antimicrobial coatings, and more specifically those that explore the antimicrobial properties of silver ions, the authors must emphasize the novelty of their study, both in abstract and introduction.

- Some figures are not very sharp and figure 3 it is missing some images.

Author Response

Comment 1.1.

- According to the data that the authors present in figure 1, the number of publications concerning antibacterial nanoparticles and antibacterial silver nanoparticles is diminishing. That trend has a reason, which is mainly associated with the development of bacterial resistance to metallic ions and nanoparticles, similarly to what happens with antibiotics. This aspect must be addressed by the authors in the introduction.

Reply 1.1.

We thank the Reviewer for careful reading of the manuscript and important information.

Unfortunately, new data on the development of bacterial resistance against NPs of metals and metal oxides have recently appeared. Bacterial defense mechanisms include increased expression of extracellular matrix molecules (flagellin) to inactivate NPs, release of pigments to inactivate metal ions, and activation of antioxidant defense to combat oxidative stress [1, 2]. It should be noted that these mechanisms are most likely implemented after a certain time after a single introduction of NPs [3]. We assume that the constant dosed release of new portions of NPs from a material during their using can be a way to reduce bacterial resistance against NPs. Composite materials based on polymers and NPs are capable of controlled release of NPs.

Necessary information has been added to the introduction.

After rereading of the article Figures 1 and 2 were removed to prevent of overload the introduction.

Comment 1.2.

- Due to the high number of publications on the general subject of antimicrobial coatings, and more specifically those that explore the antimicrobial properties of silver ions, the authors must emphasize the novelty of their study, both in abstract and introduction.

Reply 1.2.

We thank the Reviewer for important comment.

In present study we the first time created composite coating based on fluoroplast and Ag2O NPs. This clarification has been added to the introduction and abstract.

Comment 1.3.

- Some figures are not very sharp and figure 3 it is missing some images.

Reply 1.3.

We thank the Reviewer for carefully reading of the text. Figures 1a-c, 4, 7 and 8 were replaced with more clear.

  1. Niño-Martínez, N.; Salas Orozco, M.F.; Martínez-Castañón, G.-A.; Torres Méndez, F.; Ruiz, F. Molecular Mechanisms of Bacterial Resistance to Metal and Metal Oxide Nanoparticles. In Secondary Molecular Mechanisms of Bacterial Resistance to Metal and Metal Oxide Nanoparticles [Online] ed.; Editor, Ed.^Eds. 2019; p.^pp. Number of. (accessed. https://doi.org/10.3390/ijms20112808
  2. Amaro, F.; Morón, Á.; Díaz, S.; Martín-González, A.; Gutiérrez, J.C. Metallic Nanoparticles-Friends or Foes in the Battle against Antibiotic-Resistant Bacteria? Microorganisms 2021, 9 (2). https://doi.org/10.3390/microorganisms9020364
  3. Helmlinger, J.; Sengstock, C.; Groß-Heitfeld, C.; Mayer, C.; Schildhauer, T.; Köller, M.; Epple, M. Silver nanoparticles with different size and shape: equal cytotoxicity, but different antibacterial effects. RSC advances 2016, 6 (22), 18490-18501. https://doi.org/10.1039/C5RA27836H

Reviewer 2 Report

Dear authors,

The manuscript Fluoroplast doped by Ag2O nanoparticles as new antibacterial coating for meat industry presents the results on the development of a coating with Ag2O nanoparticles with antibacterial activity for the meat industry. However, the title lacks congruence with the objective, since it mentions more parameters to be determined, not only antibacterial activity.

The methodology describes the investigation of properties to prevent the formation of biofilms, but the results are not described.

If the coating is intended for the meat industry, why were no tests carried out on surfaces used in that industry?

According to what criteria were the pathogens whose antibacterial activity was to be investigated chosen? It is known that one of the important pathogens in that industry is pathogen E. coli. S. aureus and Pseudomonas are not very important. So why didn't they include pathogen E. coli?

Figure 1. is not necessary, it does not contribute anything interesting to the document.

Figure 2. Not necessary, the document is not a review but a research article.

Line 155. supernotant should be corrected to supernatant.

In Salmonella nomenclature, Typhimurium is a serotype, not a specie, so it should be written without italics and with capital letter.

Line 244. The process of dilution and preparation of the pathogen inoculum is not clear to me.

What was the initial concentration of the pathogens used in the evaluation of antibacterial activity?

Line 250. What medium was used as nutrient agar

Line 252. Separate this paragraph

Line 268. Which mouse strain was used in the cytotoxicity assay.

Figure 3. Does not appear complete and lacks image resolution (d).

The conclusion cannot assure that the coating can be used on meat industry cutting boards, because it was not evaluated on that type of material.

Kind regards

Author Response

Comments and Suggestions for Authors

Dear authors,

Comment 2.1

The manuscript Fluoroplast doped by Ag2O nanoparticles as new antibacterial coating for meat industry presents the results on the development of a coating with Ag2O nanoparticles with antibacterial activity for the meat industry. However, the title lacks congruence with the objective, since it mentions more parameters to be determined, not only antibacterial activity.

Reply 2.1

We thank Reviewer for carefully reading of the text.

The title “Fluoroplast doped by Ag2O nanoparticles as new antibacterial coating for meat industry” was corrected on “Fluoroplast doped by Ag2O nanoparticles as new repairing non-cytotoxic antibacterial coating for meat industry”

Comment 2.2

The methodology describes the investigation of properties to prevent the formation of biofilms, but the results are not described.

Reply 2.2

Antibiofilm activity can be evaluated by spectral methods of analysis or microscopy [1, 2]. We used fluorescence microscopy with double staining with SYTO®9 and propidium iodide fluorescent dyes. A significant violation of bacterial cell morphology and destruction of the biofilm were found. This clarification was added in the text. Figure 8c was replaced with more clear.

Comment 2.3

If the coating is intended for the meat industry, why were no tests carried out on surfaces used in that industry?

Reply 2.3

Teflon (polytetrafluoroethylene, PTFE, fluoroplast) is a commonly used material in the meat processing industry. In addition, fluoroplastic coatings can be used to modify the properties of work surfaces in the food industry [3]. We examined the ability of developed coating to adhere the surface of a fluoroplastic cutting board damaged during operation and restore its damage (sections 2.2 and 3.2).

Comment 2.4

According to what criteria were the pathogens whose antibacterial activity was to be investigated chosen? It is known that one of the important pathogens in that industry is pathogen E. coli. S. aureus and Pseudomonas are not very important. So why didn't they include pathogen E. coli?

Reply 2.4

We thank the Reviewer for an important comment. The study used strains provided by Laboratory of Microbiology of the Research Institute of Food Systems named after V.I. Gorbatov. These strains have previously been isolated from samples of meat products and from work surfaces at meat processing plants. There strains have high epidemiological significance [4-7]. For example, Salmonella is one of the most frequently isolated foodborne pathogens. It is a major worldwide public health concern, accounting for 93.8 million foodborne illnesses and 155,000 deaths per year [7]. Staphylococcus aureus leads to 240,000 cases of food poisoning occur per year, resulting in 1000 hospitalization only in USA [5]. Undoubtedly, some strains of E. coli (for example, E. coli O104:H4) are of high epidemiological importance [8]. However, most strains of E. coli do not pose a significant threat.

Comment 2.5

Figure 1. is not necessary, it does not contribute anything interesting to the document.

Reply 2.5

Figure 1. was removed from the text.

Comment 2.6

Figure 2. Not necessary, the document is not a review but a research article.

Reply 2.6

Figure 2. was removed from the text.

Comment 2.7

Line 155. supernotant should be corrected to supernatant.

Reply 2.7

Corrected

Comment 2.8

In Salmonella nomenclature, Typhimurium is a serotype, not a specie, so it should be written without italics and with capital letter.

Reply 2.8

The name of microorganism was corrected to “Salmonella enterica serotype Typhimurium”

Comment 2.9

Line 244. The process of dilution and preparation of the pathogen inoculum is not clear to me.

Reply 2.9

For the study, a daily broth culture of the studied microorganism with initial concentration 109 CFU/ml was diluted 100 times to final concentration 107 CFU/ml in sterile LB broth and poured into sterile test tubes (V = 2 ml). This clarification was added to the text.

Comment 2.10

What was the initial concentration of the pathogens used in the evaluation of antibacterial activity?

Reply 2.10

The initial concentration of each pathogens was 107 CFU/ml in sterile LB medium. This clarification was added to the text.

Comment 2.11

Line 250. What medium was used as nutrient agar

Reply 2.11

LB agar was used as dense nutrient medium. This clarification was added to the text.

Comment 2.12

Line 252. Separate this paragraph

Reply 2.12

Corrected

Comment 2.13

Line 268. Which mouse strain was used in the cytotoxicity assay.

Reply 2.13

BALC/b mice were used in cytotoxicity assay. This clarification was added to the text.

Comment 2.14

Figure 3. Does not appear complete and lacks image resolution (d).

Reply 2.14

Figure 3d was replaced by more clear TEM image.

Comment 2.15

The conclusion cannot assure that the coating can be used on meat industry cutting boards, because it was not evaluated on that type of material.

Reply 2.15

Teflon (polytetrafluoroethylene, PTFE, fluoroplast) is a commonly used material in the meat processing industry, including cutting board production. We used Teflon cutting board provided by meat manufacturer.

Kind regards

Submission Date

30 November 2022

Date of this review

12 Dec 2022 20:06:24

© 1996-2022 MDPI (Basel, Switzerland) unless otherwise stated

Disclaimer Terms and Conditions Privacy Policy

  1. Mombeshora, M.; Chi, G.F.; Mukanganyama, S. Antibiofilm Activity of Extract and a Compound Isolated from Triumfetta welwitschii against Pseudomonas aeruginosa. Biochemistry Research International 2021, 2021, 9946183. https://doi.org/10.1155/2021/9946183
  2. Ahamad, I.; Bano, F.; Anwer, R.; Srivastava, P.; Kumar, R.; Fatma, T. Antibiofilm Activities of Biogenic Silver Nanoparticles Against Candida albicans. Frontiers in Microbiology 2022, 12. https://doi.org/10.3389/fmicb.2021.741493
  3. Gubanova, M.I.; Kirsh, I.A.; Bannikova, O.A.; Beznaeva, O.V. Development of Anti-Adhesive Fluoroplastic Coatings for Food Technologies (In Russ.). Health, Food & Biotechnology 2020, 2 (1), 49-61. https://doi.org/10.36107/hfb.2020.i1.s293
  4. Thakur, M.; Asrani, R.K.; Patial, V. Chapter 6 - Listeria monocytogenes: A Food-Borne Pathogen Periodical Chapter 6 - Listeria monocytogenes: A Food-Borne Pathogen [Online], 2018, p. 157-192. https://www.sciencedirect.com/science/article/pii/B9780128114445000063 (accessed Date Accessed). https://doi.org/10.1016/B978-0-12-811444-5.00006-3
  5. Fetsch, A.; Johler, S. Staphylococcus aureus as a Foodborne Pathogen. Current Clinical Microbiology Reports 2018, 5 (2), 88-96. https://doi.org/10.1007/s40588-018-0094-x
  6. Rezaloo, M.; Motalebi, A.; Mashak, Z.; Anvar, A. Prevalence, Antimicrobial Resistance, and Molecular Description of <i>Pseudomonas aeruginosa</i> Isolated from Meat and Meat Products. Journal of Food Quality 2022, 2022, 9899338. https://doi.org/10.1155/2022/9899338
  7. Eng, S.-K.; Pusparajah, P.; Ab Mutalib, N.-S.; Ser, H.-L.; Chan, K.-G.; Lee, L.-H. Salmonella: A review on pathogenesis, epidemiology and antibiotic resistance. Frontiers in Life Science 2015, 8 (3), 284-293. https://doi.org/10.1080/21553769.2015.1051243
  8. Yang, S.C.; Lin, C.H.; Aljuffali, I.A.; Fang, J.Y. Current pathogenic Escherichia coli foodborne outbreak cases and therapy development. Archives of microbiology 2017, 199 (6), 811-825. https://doi.org/10.1007/s00203-017-1393-y

Reviewer 3 Report

The authors of the manuscript present their work on antibacterial coatings based on fluoroplasts combined with Ag2O nanoparticles. The work is very well presented, the experimental section is especially detailed, which is something I wish would be a standard in scientific publications. There are only several minor errors:

In the sentence in lines 76-79 there is a verb missing.

In Figure 3 the a, b and c parts are missing.

The description on lines 381-386 seems to be duplicated.

Line 387 should probably state: "The aggregate lenght" instead of "The aggregate width".

The end of a sentence is missing in line 547.

Author Response

Comment 3.1

In the sentence in lines 76-79 there is a verb missing.

Reply 3.1

Authors thank Reviewer for carefully reading of the text.

The sentence was corrected on “Among the complications of bacterial infections were reported lesions of the gastrointestinal tract (gastritis, stomach ulcers, severe forms of diarrhea), CNS (meningitis, encephalitis), kidneys, liver, spleen, musculoskeletal system (reactive arthritis), cardiovascular system (endocarditis) and reproductive system (premature birth, stillbirth).”

Comment 3.2

In Figure 3 the a, b and c parts are missing.

Reply 3.2

Fixed.

Comment 3.3

The description on lines 381-386 seems to be duplicated.

Reply 3.3

Authors thank Reviewer for carefully reading of the text.

Duplicate information was removed.

Comment 3.4

Line 387 should probably state: "The aggregate lenght" instead of "The aggregate width".

Reply 3.4

We agree with the Reviewer and corrected this phrase.

Comment 3.5

The end of a sentence is missing in line 547.

Reply 3.5

Authors thank Reviewer for thorough reading of the manuscript.

We corrected the sentence to “Probably, the increase in cell culture density on the fluoroplast/NP Ag2O composite is due to the presence of NP Ag2O in concentration more than 2 μg/ml.”

Round 2

Reviewer 1 Report

The authors are acknowledge for taking in consideration the reviewers comments/suggestions

Reviewer 2 Report

Dear Authors

Thank you for responding to the recommendations.